# Nasopharyngeal colonization by *Streptococcus pneumoniae* in children and adults before the introduction of the 10-valent conjugate vaccine, Paraguay

**Gustavo Chamorro**[1], **Aníbal Kawabata**[1], **María da Gloria Carvalho**[2], **Fabiana C. Pimenta**[2], **Fernanda C. Lessa**[2], **Carlos Torres**[3], **María José Lerea**[3], **María Eugenia León**[1] *

1 Central Laboratory of Public Health, Ministry of Public Health and Social Welfare, Asunción, Paraguay,
2 Division of Bacterial Diseases, National Center for Immunization and Respiratory Diseases, Centers for Disease Control and Prevention, Atlanta, Georgia, United States of America, 3 National Program on Immunopreventable Diseases and Expanded Program on Immunizations, Asunción, Paraguay

* maruleonayala@hotmail.com

**Data Availability Statement:** The minimal dataset underlying the results described in the paper can

## Abstract

*Streptococcus pneumoniae* is a cause of invasive diseases such as pneumonia, meningitis, and other serious infections among children and adults in Paraguay. This study was conducted to establish *S. pneumoniae* baseline prevalence, serotype distribution, and antibiotic resistance patterns in healthy children aged 2 to 59 months and adults ≥60 years of age prior to the introduction of PCV10 in the national childhood immunization program in Paraguay. Between April and July 2012, a total of 1444 nasopharyngeal swabs were collected, 718 from children aged 2 to 59 months and 726 from adults ≥60 years of age. The pneumococcal isolation, serotyping, and antibiotic susceptibility testing were performed using standard tests. Pneumococcal colonization prevalence was 34.1% (245/718) in children and 3.3% (24/726) in adults. The most frequent pneumococcal vaccine-types (VT) detected in the children were 6B (42/245), 19F (32/245), 14 (17/245), and 23F (20/245). Carriage prevalence with PCV10 serotypes was 50.6% (124/245) and PCV13 was 59.5% (146/245). Among colonized adults, prevalence of PCV10 and PCV13 serotypes were 29.1% (7/24) and 41.6% (10/24), respectively. Colonized children were more likely to share a bedroom, have a history of respiratory infection or pneumococcal infection compared to non-colonized children. no associations were found in adults. However, no significant associations were found in children and neither in adults. Vaccine-type pneumococcal colonization was highly prevalent in children and rare in adults in Paraguay prior to vaccine introduction, supporting the introduction of PCV10 in the country in 2012. These data will be useful to evaluate the impact of PCV introduction in the country.

## Introduction

*Streptococcus pneumoniae* remains one of the most important causes of morbidity and mortality in children and adults around the world [1]. According to the World Health Organization

be found at supporting Information files and at 10.6084/m9.figshare.21089815.

**Funding:** This work was supported by grants from the Mercosur Structural Convergence Fund (FOCEM) -Mercosur, FOCEM agreement N° 03/11 Project "Research, Education and Biotechnologies Applied to Health (COF 03/11). The funders had no role in study design, data collection and analysis, decision to publish, or preparation of the manuscript. This work was largely supported with own funds from the Ministry of Public Health and Social Welfare through the PNEI / PAI and the LCSP. The funders had no role in study design, data collection and analysis, decision to publish, or preparation of the manuscript.

**Competing interests:** The autors have declared that no competing interests exist.

(WHO), of the estimated 5.83 million deaths among children <5 years of age globally in 2015, 294,000 (uncertainty range [UR], 192 000–366 000) were caused by pneumococcal infections [2]. The burden of pneumococcal disease is high in children, older adults, and those who are immunosuppressed, and most of the deaths occur in developing countries [3].

*S. pneumoniae* normally colonizes in the human upper respiratory tract. The frequency of nasopharyngeal colonization by *S. pneumoniae* varies depending on age, health, and socioeconomic status of the study population [4], but it is estimated to be between 5.0% to 75.0% [5] and the distribution of the serotypes in nasopharyngeal carriage or invasive disease varies according to geographic location [6]. Children are more commonly colonized with *S. pneumoniae* (20–50%) than adults (5–20%), and the highest colonization rates are found among children less than five years of age, which corresponds precisely with the highest incidence of pneumococcal disease [7]. The prevalence of pneumococcal colonization increases in the first years of life, reaching a peak of approximately 50.0% to 80.0% in children aged 2 to 3 years and decreasing thereafter until stabilizing at 5.0% to 10.0% in children >10 years of age [8]. Children have been suggested to be the source of transmission in the household and also be involved more in community transmission than adults [9].

Unlike the amount of information available on colonization in children, less is known about pneumococcal colonization in older adults, especially in low- and middle-income countries even though studies have reported that it is lower [7] than the rate found in children [10]. Colonization is a precondition for pneumococcal disease [11].

Paraguay introduced the 10-valent pneumococcal conjugate vaccine (PCV10) in 2012. The introduction of PCV10 reduces vaccine-type pneumococcal colonization in vaccinated individuals, which leads to a decrease in transmission and indirect protection of unvaccinated individuals (herd immunity). However, it can lead to a gradual increase in non-vaccine serotypes [12–14].

We conducted a cross-sectional pneumococcal colonization survey among children and adults to better understand pneumococcal colonization prevalence, serotype distribution, and antibiotic resistance patterns to establish baseline pneumococcal carrier status.

## Materials and methods

### Setting, design and study period

As part of the national surveillance program for meningitis and pneumonia, a cross-sectional study was carried out, whose study period was from 2012 to 2020.

### Study population and samples

We enrolled all children aged 2 to 59 months and adults ≥60 years of age who met the inclusion criteria, from April through July 2012 across 10 hospitals in urban area, from outpatient corresponding to the following departments: Central (Regional Hospital of Luque in Luque City, Acosta Ñú General Pediatric Hospital in San Lorenzo City, Distrital Hospital Ñemby in Ñemby City); Amambay (Regional Hospital Pedro Juan Caballero in Pedro Juan Caballero City); Itapúa (Regional Hospital Encarnación in Encarnación City); Caaguazú (Regional Hospital Coronel Oviedo in Coronel Oviedo City); Alto Paraná (Regional Hospital Ciudad del Este in Ciudad del Este City); and the capital Asunción City (Barrio Obrero Hospital, San Pablo Hospital). A nasopharyngeal (NP) swab was collected for each participant after written consent from parent (children) or the adult. In December 2012, the Ministry of Health and Social Welfare of Paraguay introduced PCV10 to the regular vaccination schedule of the Expanded Program on Immunizations, using the three-dose schedule (Scheme 2 + 1): two

primary doses in children <1 year (2 to 11 months) and a booster at one year of age (12 months or 6 months after the second dose).

### Inclusion criteria

Individuals presented to outpatient clinics for conditions other than an acute illness and who did not have fever or immune disorders.

### Exclusion criteria

Documentation of any dose of pneumococcal vaccine or incomplete information to accurately determine immunization status (PCV7 was already licensed in Paraguay and available in the private sector).

### Isolation, identification, and serotyping

For each participant, a trained staff collected specimen from the nasopharynx using calcium alginate swabs (Fisher Brand). The swab was placed in a cryotube containing STGG (skim milk, tryptone, glucose, glycerin) media, vortexed, and stored at -80˚C [15]. A total of 10% of the NP swabs (random sample) were cultured upon arrival at the Central Laboratory of Public Health for initial quality control using the methodology described below.

All the NP swabs were stored at -80˚C and shipped in October 2014 to the *Streptococcus* Laboratory at the U.S. Centers for Disease Control and Prevention (CDC) in dry ice for processing. Pneumococci were isolated from STGG-NP by inoculating 200 µL into 5 mL Todd-Hewitt broth with 0.5% yeast extract plus 1 mL of rabbit serum (enrichment) followed by colony isolation on 5% blood agar plates [16]. Suspect alpha-hemolytic colonies were confirmed by standard microbiological tests, including colony morphology, optochin susceptibility, and bile solubility tests [17].

The serotypes of the pneumococcal isolates were identified by Quellung reaction and multiplex PCR [18]. The serotypes were classified as vaccine type (VT): PCV10 serotypes (1, 4, 5, 6B, 7F, 9V, 14, 18C, 19F, 23F), PCV13 serotypes (PCV10 serotypes plus serotypes 3, 6A, and 19A), PCV15 serotypes (PCV13 plus 22F and 33F), PCV20 serotypes (PCV15 plus 8, 10A, 11A, 12F, and 15B), or non-vaccine type (NVT, all other serotypes). When more than one potential pneumococcal colony type was identified, each colony morphology was tested, and all serotypes identified from the unique specimen were included in the analysis.

### Antibiotic susceptibility

Pneumococcal isolates were tested for susceptibility to commonly used antibiotics [amoxicillin/clavulanic acid, azithromycin, cefepime, cefotaxime, ceftriaxone, cefuroxime, chloramphenicol, clindamycin, daptomycin, ertapenem, erythromycin, levofloxacin, linezolid, penicillin, meropenem, moxifloxacin, tetracycline, trimethoprim/sulfamethoxazole, vancomycin] using broth microdilution method (Trek Diagnostics, Cleveland OH) at the *Streptococcus* Laboratory at CDC Atlanta in December 2020. Isolates were classified as susceptible, intermediate, or resistant based on the non-parenteral breakpoints of the 2020 Clinical and Laboratory Standards Institute (CLSI) guidelines [19].

### Risk factors

The following risk factors were considered in healthy children aged 2 to 59 months of age and adults ≥ 60 years: sharing a room, attending daycare, infectious processes of the upper respiratory tract in the last month, congenital disease, history of a previous episode of pneumonia,

and if the person enrolled (≥60 years old) smokes or if there are smokers in the house. These data were collected in a standardized form.

## Statistical analysis

The data obtained in the investigation were entered into a database in Excel format. Statistical analyzes were performed using STATA software [20], version 11.0 (Stata Corp, College Station, TX). Initially, a general analysis of the age of the population included in the sample was made, using the measures of central tendency and dispersion.

Demographic, clinical, and socio-economic characteristics were compared using chi-square for categorical variables and t-test for continuous variable. Two-sided P value of 0.05 was considered statistically significant. Multivariable analysis, was performed to evaluate independent risk factors associated with pneumococcal colonization. Separate multivariable models were constructed for each outcome using automated stepwise logistic regression (congenital disease, pneumonia and presence of smokers in the household). Variables with P values< 0.30 in univariable analysis were included as candidate variables for the model. Multivariable analyses was not conducted if univariable analyses showed <2 variables with P<0.30 for the outcome of interest.

## Ethical aspects

This study was approved by the research ethics committee of the Central Laboratory of Public Health (Cod.: 014/28112011). CDC provided technical assistance and tested coded specimens, and therefore was considered not engaged in research. A non-disclosure agreement was signed between CDC and the Paraguay MHWS Central Public Health Laboratory. Informed consent was obtained from children's guardians and from adults. The NP swabs and survey questionnaires were numerically coded and processed anonymously. The data were stored in a database and use restricted for this analysis.

## Results

### Demographic data

A total of 1444 participants who attended the 10 health services were included in this study, and 97 individuals were excluded due to sample contamination or missing demographic data. Half of the participants were children [718 (49,72%)], of those, 350 (48.74%) were male. Most of the children were from Central (n = 362), Asunción (n = 135), and Caaguazú (n = 93). Among the 726 adults, 472 (65.0%) were female, 362 (49.8%) were 60–69 years old, 283 (39.0%) were 70–79 year old, and 81 (11.2%) were ≥80 years of age. The majority of the adults were from Central (n = 299), Asunción (n = 188), Alto Parana (n = 59). The overall prevalence of pneumococcal colonization in children was 34.1% (95% CI 31.69–36.65) and in adults was 3.3% (95% CI 2.46–4.38). The prevalence of pneumococcal colonization was highest among children aged 2–11 months and lowest among those 36–59 months old (Table 1).

Pneumococcal colonization prevalence was similar between gender in both children and adults. Children from the Department of Amambay presented the highest pneumococcal carriage rate with 52.7% (19/36), followed by the children from the Department of Caaguazú 41.9% (39/93) and the Central Department with 30.7% (111/362).

Among the 245 children colonized with pneumococci, 38 serotypes were identified. Among the 24 adults colonized with pneumococci, 15 serotypes were identified (Fig 1). The VT pneumococcal serotypes in children accounted for 49.4% (PCV10), 57.9% (PCV13), 60.0%

**Table 1. Prevalence of pneumococcal colonization of the study population by age group (n = 1444).**

| Variable | Category | 2–59 months | | | | Category | ≥ 60 years | | | |
|---|---|---|---|---|---|---|---|---|---|---|
| | | Total tested | Pneumococcal carriers | % | IC95% | | Total tested | Pneumococcal carriers | % | IC95% |
| Age group | 2–11 months | 88 | 44 | 50.0 | 47.39–52.61 | 60–69 years | 362 | 13 | 4.0 | 2.70–4.69 |
| | 12–23 months | 196 | 70 | 35.7 | 33.26–38.27 | 70–79 years | 283 | 9 | 3.2 | 2.34–4.23 |
| | 24–35 months | 147 | 51 | 34.7 | 32.23–37.21 | ≥ 80 years | 81 | 2 | 2.5 | 1.75–3.43 |
| | 36–59 months | 287 | 80 | 27.9 | 25.54–30.22 | | | | | |
| Gender | Male | 350 | 123 | 32.6 | 31.60–38.71 | Male | 274 | 9 | 3.3 | 2.13–4.88 |
| | Female | 368 | 122 | 31.1 | 29.71–36.72 | Female | 452 | 15 | 3,3 | 2.13–4.88 |
| Department | Asunción (Capital) | 135 | 49 | 36.3 | 32.82–39.98 | Asunción (Capital) | 188 | 8 | 4,3 | 2.92–6.01 |
| | Central | 362 | 111 | 30.7 | 27.28–34.16 | Central | 299 | 12 | 4,0 | 2.69–5.68 |
| | Alto Paraná | 55 | 15 | 27.3 | 24.07–30.71 | Alto Paraná | 59 | 0 | 0 | |
| | Itapúa | 22 | 7 | 31.8 | 28.36–35.30 | Itapúa | 20 | 1 | 5 | 3.49–6.79 |
| | Amambay | 36 | 19 | 52.7 | 49.06–56.49 | Amambay | 35 | 1 | 2.9 | 1.79–4.38 |
| | Caaguazú | 93 | 39 | 41.9 | 38.28–45.63 | Caaguazú | 120 | 2 | 1.7 | 0.85–2.87 |
| | Others[1] | 15 | 5 | 33.3 | 29.84–36.87 | Others[1] | 5 | 0 | 0 | |

**Others[1]** San Pedro (n = 3), Cordillera (n = 11), Misiones (n = 2), Paraguarí (n = 1), Presidente Hayes (n = 2), Boquerón (n = 1).

(PCV15), and 70.2% (PCV20). The most frequent serotypes were 6B, 19F, 23F, and 14. The most frequent non-PCV13 serotypes in children in this study were 15B, 11A, 16F, 23B, and 13.

Children who shared a bedroom had 2.3 times increased odds of being a pneumococcal carrier compared to those who did not share a bedroom (35% vs.18%, p = 0.03) (Table 2). However, this association was no longer significant after adjusting for share room (0.072), smokers present in the household (p = 0.194), congenital condition (p = 0.569), and pneumonia (p = 0.118). In adults, no statistically significant associations with pneumococcal carriage were observed (Table 3).

All *S. pneumoniae* isolates were susceptible to levofloxacin, ertapenem, moxifloxacin, linezolid, and vancomycin. Out of 269 pneumococcal isolates, 64 (23.9%) were resistant to trimethoprim/sulfamethoxazole, 50 (18.7%) to tetracycline, 49 (18.3%) to azithromycin and erythromycin, 32 (11.9%) to cefuroxime, 26 (9.7%) to clindamycin, and 10 (3.7%) to chloramphenicol. Some pneumococcal isolates presented intermediate susceptibility to cefepime 17 (6.3%), penicillin 10 (3.7%), and ceftriaxone 14 (5.2%). In children, of the 245 pneumococcal isolates, 19 (7.7%) were resistant to erythromycin-trimethoprim/sulfamethoxazole-tetracycline, 18 (7.3%) were resistant to erythromycin-trimethoprim/sulfamethoxazole-tetracycline-cefuroxime, 18 (7.3%) were resistant to erythromycin-clindamycin-trimethoprim/sulfamethoxazole-tetracycline, and 6 (2.4%) were resistant to erythromycin-trimethoprim/sulfamethoxazole-tetracycline-chloranphenicol. In adults, of the 24 pneumococcal isolates, 3 (12.5%) were

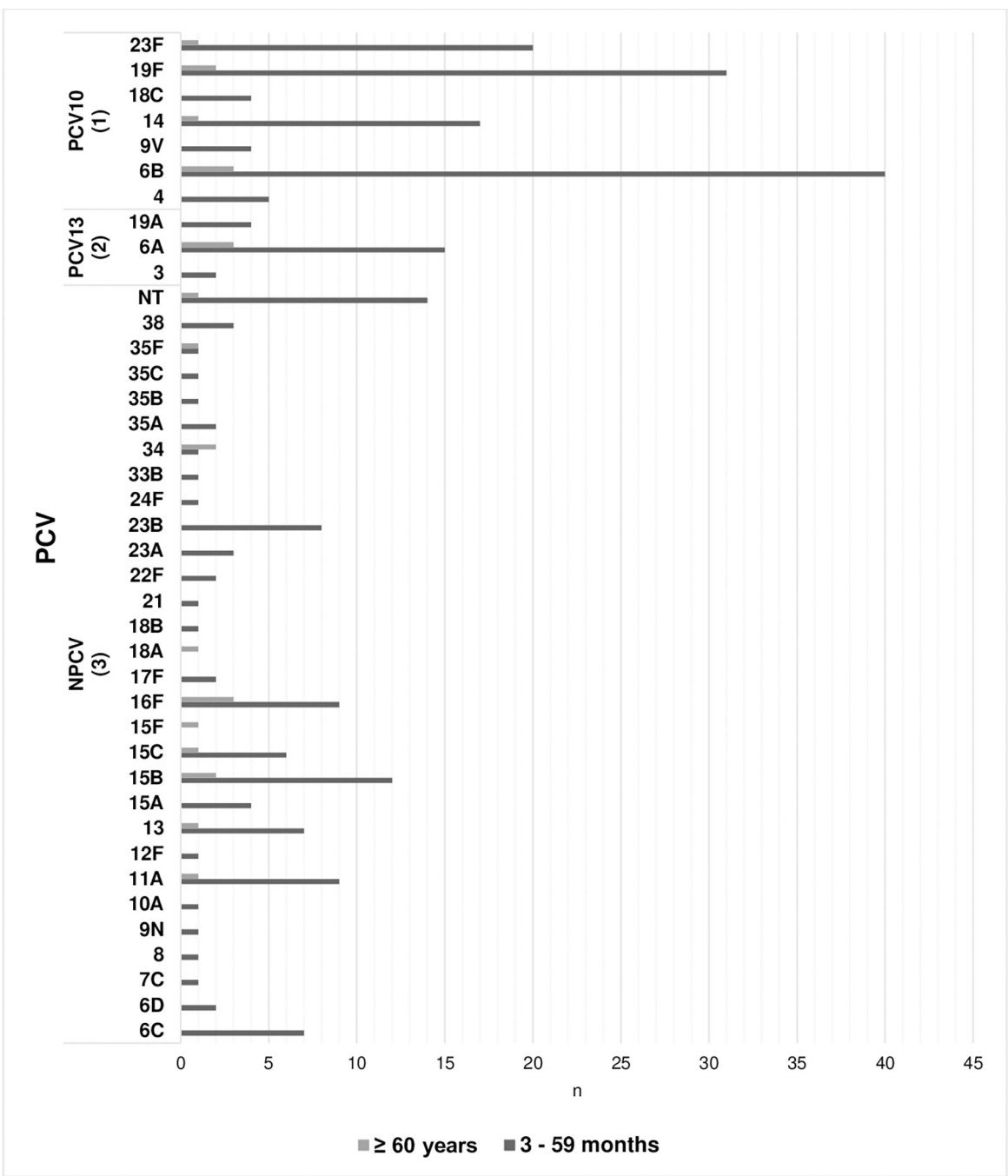

**Fig 1. Distribution of pneumococcal serotypes by age group in Paraguay (n = 269).** [1]PCV10: 10 valent Pneumococcal conjugate vaccine. [2]PCV13: 13 valent Pneumococcal conjugate vaccine. [3]NPCV: Serotypes not included in PCV10 and PCV13. NT: Non typeable. n: Isolates numbers.

resistant to erythromycin-trimethoprim/sulfamethoxazole-tetracycline, 2 (8.3%) were resistant to erythromycin-trimethoprim/sulfamethoxazole-tetracycline-cefuroxime, and 2 (8.3%) were resistant to erythromycin-clindamycin-trimethoprim/sulfamethoxazole-tetracycline. (Table 4).

**Table 2. Association between risk factors and nasopharyngeal colonization with *S. pneumoniae*, according to age group.**

| Risk factor | Total | OR | 95% CI | *p-value* |
|---|---|---|---|---|
| **2–59 months (n = 245)** | | | | |
| Share room | 718 | 2.38 | 1.03–5.49 | ***0.036*** |
| Smokers present | 715 | 1.31 | 0.93–1.84 | *0.124* |
| Attends daycare | 547 | 1.31 | 0.93–1.84 | *0.928* |
| Congenital condition | 718 | 1.54 | 0.69–3.45 | *0.289* |
| Respiratory infection | 718 | 1.30 | 0.95–1.77 | *0.095* |
| Illness/Pneumonia | 713 | 2.18 | 0.95–5.02 | *0.060* |
| **≥ 60 years (n = 24)** | | | | |
| Share room | 726 | 0.89 | 0.39–2.08 | *0.804* |
| Smokers in the house | 726 | 1.47 | 0.76–2.69 | *0.184* |
| You smoke | 709 | 1.33 | 0.30–5.86 | *0.704* |
| Congenital condition | 726 | . | . | *0.501* |
| Respiratory infection | 726 | 1.62 | 0.66–3.99 | *0.286* |
| Illness/Pneumonia | 660 | 0.71 | 0.09–5.45 | *0.743* |

## Discussion

Before the introduction of PCV10 in Paraguay, 31.86% of children aged ≤59 months were colonized with pneumococci, and to the best of our knowledge this is the first study providing baseline data prior to PCV introduction and that might allow vaccine impact evaluation.

In this study, the VT pneumococcal serotypes were the most frequent. The prevalence of VT is similar to other pre-vaccine introduction reports from other Latin American countries. In Cuba, pneumococcal colonization was found in 21.60% of children aged >18 months and the most frequently found serotypes were 6A, 23F, 6B, 19F, and 14 [21]. The colonization rate in children from Niterói, Brazil, was 49.2% and the most prevalent serotypes were 6B, 19F, 6A, 14, 15C, and 23F [22]. Fernandez et al described a colonization of 49.7% in children <2 years of age in Paysandú, Uruguay, with predominance of serotypes 6A, 6B, 14, and 19F [23].

A study conducted in children <5 years of age from southeast Brazil before the introduction of PCV10 also detected serotypes 11A, 23B, and 16F [24].

In this study, colonization in children was 10 times higher than in adults. There are few previously published studies of colonization in adults and the rates reported were 2 to 6 times higher in children than in adults [25]. A study in Kenya found among subjects ages 0–4, 5–9, and 10–85 years, that the prevalence of pneumococcal colonization was 57.0%, 41.0%, and 6.4%, respectively [26].

Currently, the available conjugate vaccines reduce the colonization of vaccine serotypes, as well as the burden of pneumococcal disease [27].

In children, sharing a room was the only variable associated with colonization of *S. pneumoniae* in this study. A study in five cities in Argentina found that children who shared a

**Table 3. Multivariable analysis for pneumococcal colonization risk factors among children (n = 245).**

| Risk factor | OR (95% CI) | *P-adjusted* |
|---|---|---|
| Share room | 2.38 (1.03–5.49) | *0.072* |
| Smokers present | 1.31 (0.93–1.84) | *0.194* |
| Congenital condition | 1.54 (0.69–3.45) | *0.569* |
| Illness/Pneumonia | 2.18 (0.95–5.02) | *0.118* |

**Table 4. Distribution of antimicrobial resistance by serotypes.**

**2–59 months**

| Antibiotics | PCV10 | | | | | | | | | | PCV13 | | | p |
| --- | --- | --- | --- | --- | --- | --- | --- | --- | --- | --- | --- | --- | --- | --- |
| | 4 | 6B | 9V | 14 | 18C | 19F | 23F | 1 | 5 | 7F | 3 | 6A | 19A | |
| Penicillin G | 0 | 0 | 0 | 0 | 0 | 0 | 0 | 0 | 0 | 0 | 0 | 0 | 0 | 0.923 |
| Ceftriaxone | 0 | 0 | 0 | 0 | 0 | 0 | 0 | 0 | 0 | 0 | 0 | 0 | 0 | 0.504 |
| Azithromycin | 0 | 7(16.67) | 0 | 7(41.18) | 0 | 18(56.25) | 4(20.00) | 0 | 0 | 0 | 0 | 7(16.67) | 0 | 0.002 |
| Tetracycline | 0 | 6(14.29) | 0 | 7(41.18) | 1(25.00) | 16(50.00) | 5(25.00) | 0 | 0 | 0 | 1(50.00) | 0 | 0 | <0.001* |
| Erythromycin | 0 | 7(16.67) | 0 | 7(41.18) | 0 | 18(56.25) | 4(20.00) | 0 | 0 | 0 | 0 | 7(4375) | 0 | 0.002 |
| TMP–SMX | 0 | 8(19.05) | 3(75.00) | 11(64.71) | 0 | 18(56.25) | 7(35.00) | 0 | 0 | 0 | 0 | 1(6.25) | 0 | 0.001 |
| Clindamycin | 0 | 6(14.29) | 1(25.00) | 1(5.88) | 1(25.00) | 16(50.00) | 0 | 0 | 0 | 0 | 0 | 0 | 0 | 0.221 |
| Chloramphenicol | 0 | 2(4.76) | 0 | 0 | 0 | 1(3.13) | 2(10.00) | 0 | 0 | 0 | 0 | 0 | 0 | 0.030 |
| AMC | 0 | 0 | 0 | 0 | 0 | 1(3.13) | 0 | 0 | 0 | 0 | 0 | 0 | 0 | 1.000 |
| Cefuroxime | 0 | 2(4.76) | 0 | 4(23.53) | 0 | 16(50.00) | 3(15.00) | 0 | 0 | 0 | 0 | 0 | 0 | 0.523 |

**≥ 60 years**

| Antibiotics | 4 | 6B | 9V | 14 | 18C | 19F | 23F | 1 | 5 | 7F | 3 | 6A | 19A | p |
| --- | --- | --- | --- | --- | --- | --- | --- | --- | --- | --- | --- | --- | --- | --- |
| Penicillin G | 0 | 0 | 0 | 0 | 0 | 0 | 0 | 0 | 0 | 0 | 0 | 0 | 0 | - |
| Ceftriaxone | 0 | 0 | 0 | 0 | 0 | 0 | 0 | 0 | 0 | 0 | 0 | 0 | 0 | - |
| Azithromycin | 0 | 0 | 0 | 0 | 0 | 2(100) | 0 | 0 | 0 | 0 | 0 | 2(66.67) | 0 | 0.144 |
| Tetracycline | 0 | 0 | 0 | 1(100) | 0 | 2(100) | 0 | 0 | 0 | 0 | 0 | 1(33.33) | 0 | 0.107 |
| Erythromycin | 0 | 0 | 0 | 0 | 0 | 2(100) | 0 | 0 | 0 | 0 | 0 | 2(66.67) | 0 | 0.144 |
| TMP–SMX | 0 | 0 | 0 | 0 | 0 | 2(100) | 0 | 0 | 0 | 0 | 0 | 1(33.33) | 0 | 0.648 |
| Clindamycin | 0 | 0 | 0 | 0 | 0 | 0 | 0 | 0 | 0 | 0 | 0 | 1(100) | 0 | 0.823 |
| Chloramphenicol | 0 | 0 | 0 | 0 | 0 | 0 | 0 | 0 | 0 | 0 | 0 | 0 | 0 | 0.904 |
| AMC | 0 | 0 | 0 | 0 | 0 | 0 | 0 | 0 | 0 | 0 | 0 | 0 | 0 | - |
| Cefuroxime | 0 | 0 | 0 | 0 | 0 | 2(100) | 0 | 0 | 0 | 0 | 0 | 2(66.67) | 0 | 0.144 |

TMP–SMX: Trimethoprim-sulfamethoxazole.

AMC: Amoxicillin/Clavulanic Acid.

room with 3 or more people had an increased risk of colonization [28]. Unlike, another study from South Africa found a high prevalence of *S. pneumoniae* in patients with respiratory conditions [29]. Several studies have reported the intrafamilial spread of *S. pneumoniae* [30,31].

In this study, no association was found between attending daycare centers and colonization. However, the risk of nasopharyngeal colonization by potential pathogens, such as *S. pneumoniae*, in healthy children is higher in children who regularly attend daycare centers than in those who are cared for at home as has previously been documented [32]. This higher incidence could be explained by the characteristics and environmental conditions that exist in these institutions for children, as well as by the age of the population group. The prevalence of carriage is independent of geographical region but strongly associated with accumulated risk factors, such as young age, high-density living conditions, and poor health conditions [33,34].

In adults, no significant differences were observed between colonized and non-colonized subjects by age, gender, presence of comorbidities, vaccination status, previous respiratory infection, previous episode of pneumonia, and smokers. In Alaska overcrowding in households was associated with increased colonization at all ages, including adults. Also, the prevalence of colonization increased with the increasing number of occupants of the home and decreased with the increasing number of rooms in a home [35]. The presence of children in a home was associated with a higher prevalence of colonization for all ages, reflecting the importance of young children as potential transmitters of pneumococcus [36,37].

Our data indicated that the rates of trimethoprim-sulfamethoxazole, tetracycline, erythromycin, azithromycin, cefuroxime, and chloramphenicol resistance among isolates were found more frequently in the serotypes included in PCV10than in non-vaccine serotypes. In addition, no resistance to penicillin or third-generation cephalosporin was found. Unlike another similar study, no penicillin-resistant isolates were found [4]. High antibiotic resistance rates in *S. pneumoniae* may facilitate transmission of this pathogen among young children [38].

Pneumococci resistant to the different antibiotics were detected mainly in vaccine serotypes. The reduced susceptibility may be especially related to the circulation of vaccine serotypes typically associated with antimicrobial resistance, such as 14, 19F, and 23F [39]. PCVs can reduce antibiotic-resistant infections through direct reductions in the presence of vaccine serotypes and thus decrease the use of antibiotics [40].

Our study has some limitations. First, the data evaluated on the behaviors were based on self-reported data and could not be verified, but precautions were taken to minimize bias by cross questions. Second, although the results of this study show the period prior to vaccination in Paraguay, the long storage time of the samples prior to processing and then the analysis time, could not provide pre-vaccine information for decision-making. This study could be a baseline for future cross-sectional studies aimed at monitoring changes in the prevalence of serotype in the same population.

## Conclusion

Serotypes found among pneumococci in the nasopharynx include the most common serotypes that caused invasive disease in Paraguay in the years prior to vaccine introduction (2010–2012) [41]. In addition, in this study, strains with resistance to antimicrobials were found more frequently in serotypes included in PCV10.The data on isolates from nasopharyngeal colonization can provide relevant information on the potential burden of pneumococcal disease and are important for understanding circulation before vaccination programs are implemented as well as for evaluating the impact after vaccine introduction in the country.

## Supporting information

**S1 Table. Pneumococcal carriage dataset Paraguay.**
(XLSX)

## Author Contributions

**Conceptualization:** Gustavo Chamorro, Aníbal Kawabata, María da Gloria Carvalho, Fabiana C. Pimenta, María Eugenia León.

**Data curation:** Aníbal Kawabata, María Eugenia León.

**Formal analysis:** Aníbal Kawabata, María da Gloria Carvalho, Fabiana C. Pimenta, María Eugenia León.

**Funding acquisition:** Gustavo Chamorro.

**Investigation:** Gustavo Chamorro, Aníbal Kawabata, María da Gloria Carvalho, Fabiana C. Pimenta, Fernanda C. Lessa, Carlos Torres, María José Lerea, María Eugenia León.

**Methodology:** Gustavo Chamorro, Aníbal Kawabata, María da Gloria Carvalho, Fabiana C. Pimenta, Fernanda C. Lessa, María Eugenia León.

**Project administration:** Gustavo Chamorro.

**Resources:** Gustavo Chamorro, Carlos Torres, María José Lerea.

**Software:** Fernanda C. Lessa.

**Supervision:** María Eugenia León.

**Validation:** Gustavo Chamorro, Aníbal Kawabata, María da Gloria Carvalho, Fabiana C. Pimenta, Fernanda C. Lessa, Carlos Torres, María José Lerea, María Eugenia León.

**Visualization:** Gustavo Chamorro, Aníbal Kawabata, María da Gloria Carvalho, Fabiana C. Pimenta, Fernanda C. Lessa, María Eugenia León.

**Writing – original draft:** Gustavo Chamorro, Aníbal Kawabata, Fabiana C. Pimenta, María Eugenia León.

**Writing – review & editing:** Gustavo Chamorro, Aníbal Kawabata, María da Gloria Carvalho, Fabiana C. Pimenta, Fernanda C. Lessa, María Eugenia León.

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
