## [Decision Letter · Decision Letter 0]

28 Aug 2022

PONE-D-22-20913Nasopharyngeal colonization by Streptococcus pneumoniae in children and adults before the introduction of the 10-valent conjugate vaccine, ParaguayPLOS ONE

Dear Dr. León Ayala,

Thank you for submitting your manuscript to PLOS ONE. After careful consideration, we feel that it has merit but does not fully meet PLOS ONE’s publication criteria as it currently stands. Therefore, we invite you to submit a revised version of the manuscript that addresses the points raised during the review process.

We look forward to receiving your revised manuscript.

Kind regards,

Arghya Das, MD

Academic Editor

PLOS ONE

Journal Requirements:

Additional Editor Comments:

The study period is not clear from the text as the authors have mentioned it to be between June and December 2012 in abstract while in materials and methods it has been mentioned to be from April through December 2012.

The authors need to clarify why the exclusion criteria includes any documented dose of pneumococcal vaccine given the fact that PCV-10 was introduced in December 2012 and the study period is before the introduction of PCV-10.

The authors need to clarify the use of CLSI 2020 guidelines for the study done in the year 2012.

Reviewers' comments:

Reviewer's Responses to Questions

**Comments to the Author**

1. Is the manuscript technically sound, and do the data support the conclusions?

Reviewer #1: Partly

Reviewer #2: Yes

2. Has the statistical analysis been performed appropriately and rigorously? 

Reviewer #1: No

Reviewer #2: Yes

3. Have the authors made all data underlying the findings in their manuscript fully available?

Reviewer #1: No

Reviewer #2: Yes

4. Is the manuscript presented in an intelligible fashion and written in standard English?

Reviewer #1: Yes

Reviewer #2: Yes

5. Review Comments to the Author

Reviewer #1: The authors of the present manuscript intends to find S. pneumoniae baseline prevalence, serotype distribution, and antibiotic resistance patterns in healthy children aged 2 to 59 months and adults ≥60 years of age prior to the introduction of PCV10 in the national childhood immunization program in Paraguay. Although the study seems interesting at the first glance, there some key issues related to the selection of subjects and statistical analyses which need to be addressed by the authors. Moreover, the study was conducted almost a decade ago. It is not clear whether the study is part of an ongoing surveillance program or a standalone project. Without the follow up data after introduction of the PCV-10, the findings of the study may not be impactful.

Abstract

Lines 17-19: Colonized children were more likely to share a bedroom (P=0.03) and have a history of respiratory infection (P=0.095) or pneumococcal infection (P=0.06) compared to non-colonized children,….

Comment: The above statistics are not worth mentioning because for variables like history of respiratory infection and pneumococcal infection, the p values are higher that is assumed significant for the present study (i.e. <0.05). Moreover, sharing of bedroom has not been found significant (as mentioned under Results) after adjusting for congenital disease, pneumonia, and smokers present in the household.

Introduction

Lines 30-32: According to the World Health Organization (WHO), of the estimated 5.83 million deaths among children <5 years of age globally in 2015, 294,000 were caused by pneumococcal infections.

Comment: Please add the uncertainty interval within parenthesis after the estimated figure of 294,000.

Lines 32-34: The burden of pneumococcal disease is high in children, older adults, and those who are immunosuppressed, and it occurs in high-, low-, and middle-income countries.

Comment: The authors may consider slightly modifying the later part of the above sentence.

‘…..and most of the deaths occur in developing countries’ instead of ‘….and it occurs in high-, low-, and middle-income countries’.

Reference- https://www.cdc.gov/pneumococcal/global.html

Lines 39-42: Children are more commonly colonized with S. pneumoniae than adults, and the highest colonization rates are found among children less than five years of age, which corresponds precisely with the highest incidence of pneumococcal disease.

Comment: The statistical estimate for carriage related prevalence among both children and adults from reference no. 07 by Brooks et al. could be incorporated for the comparative depiction between the mentioned groups.

Lines 45-46: Children have been suggested to be the source of transmission in the household.

Comment: Although true, the above statement seems incomplete without mentioning the other findings from the cited literature. Like, children are also found to be involved more in community transmission than adults.

Materials and methods

Lines 63 to 76: We enrolled all children aged 2 to 59 months and adults ≥60 years of age………………………… through the Ministry of Health and Social Welfare in the health units and immunization centers in Paraguay.

Comment: The study design is poorly understood. The authors should make a flow diagram for allowing better understanding of the selection of subjects.

There are several queries which need to be clarified.

As per the title and aim of the present manuscript the authors intend to demonstrate the pneumococcal colonization prevalence, serotype, distribution, etc. before the introduction of PCV-10 vaccine. The study subjects were included from April to December 2012. As per the authors, the PCV 10 was introduced in December 2012.

However, as per a previously published study, PCV10 has been introduced much earlier than the study period.

Kieninger MP, Caballero EG, Sosa AA, Amarilla CT, Jáuregui B, Janusz CB, Clark AD, Castellanos RM. Cost-effectiveness analysis of pneumococcal conjugate vaccine introduction in Paraguay. Vaccine. 2015 May 7;33 Suppl 1:A143-53.

The relevance of the following sentence is not at all understood. “PCV10 was introduced for all eligible children (two primary doses in children <1 year and one booster at one year of age, in December 2012”

The PCV-10 vaccination schedule followed in Paraguay can be incorporated as supplementary file. If the vaccine has been introduced in December 2012 for the first time, how can the booster be considered for few children at the same time.

Lines 81-82: Documentation of any dose of pneumococcal vaccine or incomplete information to accurately determine immunization status.

Comment: Some of the important exclusion criteria have not been considered. For example, recent administration of antibiotics which might influence the S. pneumoniae colonizations. Please justify.

Line 97: Pneumococcal isolates were serotyped by Quellung reaction and multiplex PCR.

Comment: Modify the above sentence like “The serotypes of the pneumococcal isolates were identified by Quellung reaction and multiplex PCR”. The methodology of multiplex PCR should be mentioned or referenced.

Lines 106-114: Pneumococcal isolates were tested for susceptibility…… breakpoints of the 2020 Clinical and Laboratory Standards Institute (CLSI) guidelines.

Comment: The reviewer feels curious to know as when the susceptibility testing was actually performed. Because the samples were collected in 2012 and the interpretation of the test was done as per the CLSI guidelines published in 2020. Moreover, was susceptibility testing individually for all morphotypes of the colonies were performed if more than one colony type was isolated (as it is done for serotyping)?

Lines 170-173: Children who shared a bedroom had 2.3 times increased odds of being a pneumococcal carrier compared to those who did not share a bedroom (35% vs.18%, p=0.03). However, this association was no longer significant after adjusting for congenital disease, pneumonia, and smokers present in the household.

Comment: The adjusted p values could not be found in the Table 02.

How was the adjustment carried out? Was a multivariate analysis performed? More details on the statistical analyses are absent under Materials and Methods.

Lines 187-195: In children, multidrug resistance (MDR) was found in 19/245 (7.7%) of……….. and 2/24 (8.3%) of erythromycin-clindamycin-trimethoprim/sulfamethoxazole-tetracycline.

Comment: Please consider rephrasing of the two sentences mentioned in the above paragraph. It will be better if the authors present the susceptibility profile in form of graphs (showing susceptibility/resistance proportions) separately in children and adults. The above description of MDR (presumably as at least one member of ≥ 3 classes of antibiotics) seems to be complex. The authors should find easier way of depiction of the above statistics either in tables or figures.

Lines 196-198: The serotypes found most frequently with resistance to trimethoprim/sulfamethoxazole, tetracycline, erythromycin, cefuroxime, and clindamycin were 19F, 14, and 6B.

Comment: It is advisable to provide detailed statistical information before mentioning the above statement.

General comments:

The manuscript should be checked and edited by a native English speaker.

The collected data is almost a decade old. How will the authors justify the importance of the findings in recent time (given the colonization pattern have been likely changed over the past decade)? The reviewer is of the opinion that the authors may conduct a follow up surveillance to depict the changes in the colonization after 10 years of PCV-10 introduction in Paraguay.

Reviewer #2: The manuscript is written in crisp language and focused to answer the research question. But this study should be followed by another follow up study to assess the response of nationwide PCV 10 vaccination on nasopharyngeal colonization of streptococcus pneumoniae.

6. PLOS authors have the option to publish the peer review history of their article (what does this mean?). If published, this will include your full peer review and any attached files.

Reviewer #1: **Yes: **Anwita Mishra, MD

Reviewer #2: **Yes: **Saikat Mondal

---

## [Author Response · Author response to Decision Letter 0]

3 Nov 2022

Letter to Academic Editor and Revieweres

Respectfully, all the changes made to the manuscript and the pertinent clarifications are detailed below.

Journal Requirements:

Reponse:

We have reviewed the PLOS ONE style requirements.

Reponse:

We have included the title page at the beginning of the manuscript file, listing all authors and affiliations.

Reponse: 

This research has not received any specific scholarship from public sector agencies of the Ministry of Public Health and Social Welfare, PNEI-PAI or the LCSP. Therefore they do not have a grant number and we have decided to withdraw this information.

Reponse:

We have uploaded the minimum set of data underlying your study as supporting data files.

Also, the minimal dataset underlying the results described in the manuscript can be found at: 10.6084/m9.figshare.21089815 (The DOI becomes active when the item is published).

Additional Editor Comments:

The study period is not clear from the text as the authors have mentioned it to be between June and December 2012 in abstract while in materials and methods it has been mentioned to be from April through December 2012.

Reponse: The samples were collected from april to july 2012.

The authors need to clarify why the exclusion criteria includes any documented dose of pneumococcal vaccine given the fact that PCV-10 was introduced in December 2012 and the study period is before the introduction of PCV-10.

Reponse: The pneumococcal vaccine data was collected because the PCV7 was probably already licensed in Paraguay and available in the private sector.

The authors need to clarify the use of CLSI 2020 guidelines for the study done in the year 2012.

Reponse: The pneumococcal isolation was done and the isolates stored. However, the MIC testing was performed only in 2020 at CDC.

Responses to Reviewer #1

1. The authors of the present manuscript intends to find S. pneumoniae baseline prevalence, serotype distribution, and antibiotic resistance patterns in healthy children aged 2 to 59 months and adults ≥60 years of age prior to the introduction of PCV10 in the national childhood immunization program in Paraguay. Although the study seems interesting at the first glance, there some key issues related to the selection of subjects and statistical analyses which need to be addressed by the authors. Moreover, the study was conducted almost a decade ago. It is not clear whether the study is part of an ongoing surveillance program or a standalone project. Without the follow up data after introduction of the PCV-10, the findings of the study may not be impactful.

Reponse: 

Although it was planned to carry out the post-vaccination study, unfortunately, for financial reasons we have not been able to carry it out. In the same way, we believed it convenient to present the results of the pre-vaccination study, since this is the only study that has been carried out in healthy carriers in Paraguay.

Reponse: The study is part of an ongoing surveillance program. Although it was planned to carry out the post-vaccination study, unfortunately, for financial reasons we have not been able to carry it out. In the same way, we believed it convenient to present the results of the pre-vaccination study, since this is the only study that has been carried out in healthy carriers in Paraguay.

2. Abstract 

Lines 17-19: Colonized children were more likely to share a bedroom (P=0.03) and have a history of respiratory infection (P=0.095) or pneumococcal infection (P=0.06) compared to non-colonized children,….

Comment: The above statistics are not worth mentioning because for variables like history of respiratory infection and pneumococcal infection, the p values are higher that is assumed significant for the present study (i.e. <0.05). Moreover, sharing of bedroom has not been found significant (as mentioned under Results) after adjusting for congenital disease, pneumonia, and smokers present in the household.

Reponse: 

We have eliminated the statistics being as follows:

Colonized children were more likely to share a bedroom, have a history of respiratory infection or pneumococcal infection compared to non-colonized children. No associations were found in adults. However, no significant associations were found in children and neither in adults.

3. Introduction

Lines 30-32: According to the World Health Organization (WHO), of the estimated 5.83 million deaths among children <5 years of age globally in 2015, 294,000 were caused by pneumococcal infections.

Comment: Please add the uncertainty interval within parenthesis after the estimated figure of 294,000.

Reponse: 

We have added the range of uncertainty: (uncertainty range [UR], 192 000–366 000)

4. Lines 32-34: The burden of pneumococcal disease is high in children, older adults, and those who are immunosuppressed, and it occurs in high-, low-, and middle-income countries.

Comment: The authors may consider slightly modifying the later part of the above sentence.

…..and most of the deaths occur in developing countries’ instead of ‘….and it occurs in high-, low-, and middle-income countries’. Reference- https://www.cdc.gov/pneumococcal/global.html

Reponse: we have modified according to suggestion “and most of the deaths occur in developing countries”

5. Lines 39-42: Children are more commonly colonized with S. pneumoniae than adults, and the highest colonization rates are found among children less than five years of age, which corresponds precisely with the highest incidence of pneumococcal disease.

Comment: The statistical estimate for carriage related prevalence among both children and adults from reference no. 07 by Brooks et al. could be incorporated for the comparative depiction between the mentioned groups.

Reponse:

 We have added the requested prevalence, in children 20-50% and adults 5-20%.

6. Lines 45-46: Children have been suggested to be the source of transmission in the household.

Comment: Although true, the above statement seems incomplete without mentioning the other findings from the cited literature. Like, children are also found to be involved more in community transmission than adults.

Reponse: 

We have added according to the mentioned literatura and suggested “also be involved more in community transmission than adults”.

7. Materials and methods

Lines 63 to 76: We enrolled all children aged 2 to 59 months and adults ≥60 years of age………………………… through the Ministry of Health and Social Welfare in the health units and immunization centers in Paraguay.

Comment: The study design is poorly understood. The authors should make a flow diagram for allowing better understanding of the selection of subjects.

There are several queries which need to be clarified.

a) As per the title and aim of the present manuscript the authors intend to demonstrate the pneumococcal colonization prevalence, serotype, distribution, etc. before the introduction of PCV-10 vaccine. The study subjects were included from April to December 2012. As per the authors, the PCV 10 was introduced in December 2012.

However, as per a previously published study, PCV10 has been introduced much earlier than the study period.

Kieninger MP, Caballero EG, Sosa AA, Amarilla CT, Jáuregui B, Janusz CB, Clark AD, Castellanos RM. Cost-effectiveness analysis of pneumococcal conjugate vaccine introduction in Paraguay. Vaccine. 2015 May 7;33 Suppl 1:A143-53.

Reponse:

According to Castellanos RM. Cost-effectiveness analysis of pneumococcal conjugate vaccine introduction in Paraguay. Vaccine. 2015 May 7;33 Suppl 1:A143-53.

TRIVAC model version 2.0 is a tool that evaluates vaccine impact and incremental costeffectiveness ratios (ICER), providing a coherent and transparent framework for each vaccine, with comparable and standardized outcomes. Utilizing parameters, such as demography, disease burden, vaccine costs, vaccine coverage, vaccine efficacy, vaccine serotype coverage, health service utilization and costs.

In order to estimate coverage per dose in the year of introduction, EPI information on Pentavalent and MMR1 dose coverage over the last 5 years was used. 

Therefore, for this mentioned study, the real application of any PCV has not been implemented.

b)The relevance of the following sentence is not at all understood. “PCV10 was introduced for all eligible children (two primary doses in children <1 year and one booster at one year of age, in December 2012”

The PCV-10 vaccination schedule followed in Paraguay can be incorporated as supplementary file. If the vaccine has been introduced in December 2012 for the first time, how can the booster be considered for few children at the same time.

Reponse:

We have modified according to suggestion:

“In December 2012, the Ministry of Health and Social Welfare of Paraguay introduced PCV10 to the regular vaccination schedule of the Expanded Program on Immunizations, using the three-dose schedule (Scheme 2 + 1): two primary doses in children <1 year ( 2 to 11 months) and a booster at one year of age (12 months or 6 months after the second dose)”.

8. Lines 81-82: Documentation of any dose of pneumococcal vaccine or incomplete information to accurately determine immunization status.

Comment: Some of the important exclusion criteria have not been considered. For example, recent administration of antibiotics which might influence the S. pneumoniae colonizations. Please justify.

Reponse:

We agree with the reviewer that recent exposure to antibiotics may affect the colonization results either leading to lower detection of colonization or detection of resistance pneumococci. Many colonization studies do not exclude patients on antibiotics since you can still detect pneumococci among patients with recent antibiotic exposure.

9. Line 97: Pneumococcal isolates were serotyped by Quellung reaction and multiplex PCR.

Comment: Modify the above sentence like “The serotypes of the pneumococcal isolates were identified by Quellung reaction and multiplex PCR”. The methodology of multiplex PCR should be mentioned or referenced.

Reponse: 

We have modified according to suggestion: The serotypes of the pneumococcal isolates were identified by Quellung reaction and multiplex PCR.

And we added the referencePai R, Gertz RE, Beall B. Sequential multiplex PCR approach for determining capsular serotypes of Streptococcus pneumoniae isolates. J Clin Microbiol. 2006; 44(1):124-31. doi: 10.1128/JCM.44.1.124-131.2006.

10. Lines 106-114: Pneumococcal isolates were tested for susceptibility…… breakpoints of the 2020 Clinical and Laboratory Standards Institute (CLSI) guidelines.

Comment: The reviewer feels curious to know as when the susceptibility testing was actually performed. Because the samples were collected in 2012 and the interpretation of the test was done as per the CLSI guidelines published in 2020. Moreover, was susceptibility testing individually for all morphotypes of the colonies were performed if more than one colony type was isolated (as it is done for serotyping)?

Reponse: 

The pneumococcal isolation was done and the isolates stored. However, the MIC testing was performed only in 2020 at CDC.

Yes, all morphotypes of colonies with diferente serotypes were tested for antimicrobial susceptibility test by broth microdilution.

11. Lines 170-173: Children who shared a bedroom had 2.3 times increased odds of being a pneumococcal carrier compared to those who did not share a bedroom (35% vs.18%, p=0.03). However, this association was no longer significant after adjusting for congenital disease, pneumonia, and smokers present in the household.

 Comment: The adjusted p values could not be found in the Table 02.

How was the adjustment carried out? Was a multivariate analysis performed? More details on the statistical analyses are absent under Materials and Methods.

Reponse: 

We have added to methods: “Multivariable analysis, where appropriate, was performed to evaluate independent risk factors associated with pneumococcal colonization. Variables with P values< 0.30 in univariable analysis were included as candidate variables for the model. Multivariable analyses was not conducted if univariable analyses showed <2 variables with P<0.30 for the outcome of interest”.

We have modified the results:

However, this association was no longer significant after adjusting for congenital disease (p=0.569), pneumonia (p=0.118), and smokers present in the household (p=0.194). In adults, no statistically significant associations with pneumococcal carriage were observed.

Table 3. Multivariable analysis for pneumococcal colonization risk factors among children (n=245).

Risk factor OR (95% CI) P-adjusted

Share room 2.38 (1.03-5.49) 0.072

Smokers present 1.31 (0.93-1.84) 0.194

Congenital condition 1.54 (0.69-3.45) 0.569

Illness/Pneumonia 2.18 (0.95-5.02) 0.118

12. Lines 187-195: In children, multidrug resistance (MDR) was found in 19/245 (7.7%) of……….. and 2/24 (8.3%) of erythromycin-clindamycin-trimethoprim/sulfamethoxazole-tetracycline.

Comment: Please consider rephrasing of the two sentences mentioned in the above paragraph. It will be better if the authors present the susceptibility profile in form of graphs (showing susceptibility/resistance proportions) separately in children and adults. The above description of MDR (presumably as at least one member of ≥ 3 classes of antibiotics) seems to be complex. The authors should find easier way of depiction of the above statistics either in tables or figures.

Reponse: We have considered reformulating the two mentioned sentences, remaining as follows:

In children, of the 245 pneumococcal isolates, 19 (7.7%) were resistant to erythromycin-trimethoprim/sulfamethoxazole-tetracycline, 18 (7.3%) were resistant to erythromycin-trimethoprim/sulfamethoxazole-tetracycline-cefuroxime, 18 (7.3%) were resistant to erythromycin-clindamycin-trimethoprim/sulfamethoxazole-tetracycline, and 6 (2.4%) were resistant to erythromycin-trimethoprim/sulfamethoxazole-tetracycline-chloranphenicol. In adults, of the 24 pneumococcal isolates, 3 (12.5%) were resistant to erythromycin-trimethoprim/sulfamethoxazole-tetracycline, 2 (8.3%) were resistant to erythromycin-trimethoprim/sulfamethoxazole-tetracycline-cefuroxime, and 2 (8.3%) were resistant to erythromycin-clindamycin-trimethoprim/sulfamethoxazole-tetracycline.

Table 4. Distribution of antimicrobial resistance by serotypes.

2 - 59 months

Antibiotics PCV10 PCV13 p

 4 6B 9V 14 18C 19F 23F 1 5 7F 3 6A 19A 

Penicillin G 0 0 0 0 0 0 0 0 0 0 0 0 0 0.923 

Ceftriaxone 0 0 0 0 0 0 0 0 0 0 0 0 0 0.504 

Azithromycin 0 7(16.67) 0 7(41.18) 0 18(56.25) 4(20.00) 0 0 0 0 7(16.67) 0 0.002 

Tetracycline 0 6(14.29) 0 7(41.18) 1(25.00) 16(50.00) 5(25.00) 0 0 0 1(50.00) 0 0 <0.001* 

Erythromycin 0 7(16.67) 0 7(41.18) 0 18(56.25) 4(20.00) 0 0 0 0 7(4375) 0 0.002 

TMP–SMX 0 8(19.05) 3(75.00) 11(64.71) 0 18(56.25) 7(35.00) 0 0 0 0 1(6.25) 0 0.001 

Clindamycin 0 6(14.29) 1(25.00) 1(5.88) 1(25.00) 16(50.00) 0 0 0 0 0 0 0 0.221 

Chloramphenicol 0 2(4.76) 0 0 0 1(3.13) 2(10.00) 0 0 0 0 0 0 0.030 

AMC 0 0 0 0 0 1(3.13) 0 0 0 0 0 0 0 1.000 

Cefuroxime 0 2(4.76) 0 4(23.53) 0 16(50.00) 3(15.00) 0 0 0 0 0 0 0.523

≥ 60 years

Penicillin G 0 0 0 0 0 0 0 0 0 0 0 0 0 - 

Ceftriaxone 0 0 0 0 0 0 0 0 0 0 0 0 0 - 

Azithromycin 0 0 0 0 0 2(100) 0 0 0 0 0 2(66.67) 0 0.144 

Tetracycline 0 0 0 1(100) 0 2(100) 0 0 0 0 0 1(33.33) 0 0.107 

Erythromycin 0 0 0 0 0 2(100) 0 0 0 0 0 2(66.67) 0 0.144 

TMP–SMX 0 0 0 0 0 2(100) 0 0 0 0 0 1(33.33) 0 0.648 

Clindamycin 0 0 0 0 0 0 0 0 0 0 0 1(100) 0 0.823 

Chloramphenicol 0 0 0 0 0 0 0 0 0 0 0 0 0 0.904 

AMC 0 0 0 0 0 0 0 0 0 0 0 0 0 - 

Cefuroxime 0 2(66.67) 0 0 0 2(100) 0 0 0 0 0 0 0 0.144

TMP–SMX: Trimethoprim-sulfamethoxazole

AMC: Amoxicillin/Clavulanic Acid

13. The manuscript should be checked and edited by a native English speaker.

The collected data is almost a decade old. How will the authors justify the importance of the findings in recent time (given the colonization pattern have been likely changed over the past decade)? The reviewer is of the opinion that the authors may conduct a follow up surveillance to depict the changes in the colonization after 10 years of PCV-10 introduction in Paraguay.

Reponse: 

The authors agree on the importance of conducting follow-up surveillance to represent changes in colonization 10 years after the introduction of PCV-10 in Paraguay.

General comments:

The manuscript should be checked and edited by a native English speaker.

The collected data is almost a decade old. How will the authors justify the importance of the findings in recent time (given the colonization pattern have been likely changed over the past decade)? The reviewer is of the opinion that the authors may conduct a follow up surveillance to depict the changes in the colonization after 10 years of PCV-10 introduction in Paraguay.

Reponse: 

We agree with the reviewer. Although it was planned to carry out the post-vaccination study, unfortunately for economic reasons we have not been able to carry it out. Similarly, we thought it appropriate to present the results of the pre-vaccination study, since this is the only study that has been carried out in healthy carriers in Paraguay. We will do our best to carry out a postvaccinal colonization study.

Responses to Reviewer #2

Reviewer #2: The manuscript is written in crisp language and focused to answer the research question. But this study should be followed by another follow up study to assess the response of nationwide PCV 10 vaccination on nasopharyngeal colonization of streptococcus pneumoniae.

Reponse: 

We agree with the reviewer. Although it was planned to carry out the post-vaccination study, unfortunately for economic reasons we have not been able to carry it out. Similarly, we thought it appropriate to present the results of the pre-vaccination study, since this is the only study that has been carried out in healthy carriers in Paraguay.

King regards

María Eugenia León

---

## [Decision Letter · Decision Letter 1]

15 Dec 2022

PONE-D-22-20913R1Nasopharyngeal colonization by Streptococcus pneumoniae in children and adults before the introduction of the 10-valent conjugate vaccine, ParaguayPLOS ONE

Dear Dr. León Ayala,

Thank you for submitting your manuscript to PLOS ONE. After careful consideration, we feel that it has merit but does not fully meet PLOS ONE’s publication criteria as it currently stands. Therefore, we invite you to submit a revised version of the manuscript that addresses the points raised during the review process.Please submit your revised manuscript by Jan 29 2023 11:59PM. If you will need more time than this to complete your revisions, please reply to this message or contact the journal office at plosone@plos.org. Please include the following items when submitting your revised manuscript:A rebuttal letter that responds to each point raised by the academic editor and reviewer(s). You should upload this letter as a separate file labeled 'Response to Reviewers'.A marked-up copy of your manuscript that highlights changes made to the original version. You should upload this as a separate file labeled 'Revised Manuscript with Track Changes'.An unmarked version of your revised paper without tracked changes. You should upload this as a separate file labeled 'Manuscript'.If applicable, we recommend that you deposit your laboratory protocols in protocols.io to enhance the reproducibility of your results. Protocols.io assigns your protocol its own identifier (DOI) so that it can be cited independently in the future. For instructions see: https://journals.plos.org/plosone/s/submission-guidelines#loc-laboratory-protocols. Additionally, PLOS ONE offers an option for publishing peer-reviewed Lab Protocol articles, which describe protocols hosted on protocols.io. Read more information on sharing protocols at https://plos.org/protocols?utm_medium=editorial-email&utm_source=authorletters&utm_campaign=protocols.

We look forward to receiving your revised manuscript.

Kind regards,

Arghya Das, MD

Academic Editor

PLOS ONE

Journal Requirements:

Additional Editor Comments:

Multivariable analysis, where appropriate, was performed to evaluate independent risk factors associated with pneumococcal colonization.

Comment: Please specify the method that was adopted for multivariable analysis.

The serotypes found most frequently with resistance to trimethoprim/sulfamethoxazole, tetracycline, erythromycin, cefuroxime, and clindamycin were 19F, 14, and 6B (Table 4).

Comment: The inference drawn in the sentence just before Table 4 does not completely match with the statistics mentioned in the table. Hence, the above sentence should be deleted.

Reviewers' comments:

Reviewer's Responses to Questions

**Comments to the Author**

1. If the authors have adequately addressed your comments raised in a previous round of review and you feel that this manuscript is now acceptable for publication, you may indicate that here to bypass the “Comments to the Author” section, enter your conflict of interest statement in the “Confidential to Editor” section, and submit your "Accept" recommendation.

Reviewer #1: All comments have been addressed

Reviewer #3: All comments have been addressed

2. Is the manuscript technically sound, and do the data support the conclusions?

Reviewer #1: Yes

Reviewer #3: Yes

3. Has the statistical analysis been performed appropriately and rigorously? 

Reviewer #1: I Don't Know

Reviewer #3: Yes

4. Have the authors made all data underlying the findings in their manuscript fully available?

Reviewer #1: Yes

Reviewer #3: Yes

5. Is the manuscript presented in an intelligible fashion and written in standard English?

Reviewer #1: Yes

Reviewer #3: Yes

6. Review Comments to the Author

Reviewer #1: The authors have responded to the queries or comments satisfactorily. However the following needs to be incorporated in the manuscript: under material and methods the study period has still not been mentioned clearly. Kindly give specific study period details.

Reviewer #3: The authors conducted the study to establish S. pneumoniae baseline prevalence, serotype distribution, and antibiotic resistance patterns in healthy children and adults prior to the introduction of PCV10 in the national childhood

immunization program in Paraguay. And concluded with the observation of Vaccine-type pneumococcal colonization were prevalent in children and rare in adults in Paraguay prior to vaccine introduction, supporting the introduction of PCV10 in the country in 2012.

The manuscript describe a technically sound piece of scientific research with data that supports the conclusions.

7. PLOS authors have the option to publish the peer review history of their article (what does this mean?). If published, this will include your full peer review and any attached files.

Reviewer #1: **Yes: **Anwita Mishra, MD

Reviewer #3: **Yes: **Rahul Garg

---

## [Author Response · Author response to Decision Letter 1]

6 Jan 2023

Reponse to reviewers:

1. Journal Requirements:

Reponse: We have reviewed the reference list and it is complete. No new articles have been cited.

2. Multivariable analysis, where appropriate, was performed to evaluate independent risk factors associated with pneumococcal colonization.

Comment: Please specify the method that was adopted for multivariable analysis.

Reponse: The method adopted for the multivariate analysis is specified as follows:

“Multivariable analysis, was performed to evaluate independent risk factors associated with pneumococcal colonization. Separate multivariable models were constructed for each outcome using automated stepwise logistic regression (congenital disease, pneumonia and presence of smokers in the household)”.

3. The serotypes found most frequently with resistance to trimethoprim/sulfamethoxazole, tetracycline, erythromycin, cefuroxime, and clindamycin were 19F, 14, and 6B (Table 4).

Comment: The inference drawn in the sentence just before Table 4 does not completely match with the statistics mentioned in the table. Hence, the above sentence should be deleted.

Reponse: Because the inference drawn in the sentence just before Table 4 does not fully match the statistics mentioned in the table.

Deleted sentence: The serotypes found most frequently with resistance to trimethoprim/sulfamethoxazole, tetracycline, erythromycin, cefuroxime, and clindamycin were 19F, 14, and 6B.

4. Reviewer #1: The authors have responded to the queries or comments satisfactorily. However the following needs to be incorporated in the manuscript: under material and methods the study period has still not been mentioned clearly. Kindly give specific study period details.

Reponse: In order to provide specific details of the study period, the following is added in material and methods:

 Setting, design and study period

As part of the national surveillance program for meningitis and pneumonia, a cross-sectional study was carried out, whose study period was from 2012 to 2020.

5. PLOS authors have the option to publish the peer review history of their article (what does this mean?). If published, this will include your full peer review and any attached files.

Reponse: Yes, I want your identity to be public for this peer review

---

## [Decision Letter · Decision Letter 2]

8 Jan 2023

Nasopharyngeal colonization by Streptococcus pneumoniae in children and adults before the introduction of the 10-valent conjugate vaccine, Paraguay

PONE-D-22-20913R2

Dear Dr. León Ayala,

We’re pleased to inform you that your manuscript has been judged scientifically suitable for publication and will be formally accepted for publication once it meets all outstanding technical requirements.

Kind regards,

Arghya Das, MD

Academic Editor

PLOS ONE

Additional Editor Comments (optional):

Reviewers' comments:

Reviewer's Responses to Questions

**Comments to the Author**

1. If the authors have adequately addressed your comments raised in a previous round of review and you feel that this manuscript is now acceptable for publication, you may indicate that here to bypass the “Comments to the Author” section, enter your conflict of interest statement in the “Confidential to Editor” section, and submit your "Accept" recommendation.

Reviewer #1: All comments have been addressed

2. Is the manuscript technically sound, and do the data support the conclusions?

Reviewer #1: Yes

3. Has the statistical analysis been performed appropriately and rigorously? 

Reviewer #1: I Don't Know

4. Have the authors made all data underlying the findings in their manuscript fully available?

Reviewer #1: Yes

5. Is the manuscript presented in an intelligible fashion and written in standard English?

Reviewer #1: Yes

6. Review Comments to the Author

Reviewer #1: (No Response)

7. PLOS authors have the option to publish the peer review history of their article (what does this mean?). If published, this will include your full peer review and any attached files.

Reviewer #1: **Yes: **ANWITA MISHRA MD

---

## [Editor Report · Acceptance letter]

7 Feb 2023

PONE-D-22-20913R2 

Nasopharyngeal colonization by *Streptococcus pneumoniae* in children and adults before the introduction of the 10-valent conjugate vaccine, Paraguay 

Dear Dr. León:

I'm pleased to inform you that your manuscript has been deemed suitable for publication in PLOS ONE. Congratulations! Your manuscript is now with our production department. 

Kind regards, 

on behalf of

Dr. Arghya Das 

Academic Editor

PLOS ONE